# Young Single Widow, Dynamics of In-Laws Interference and Coping Mechanisms: Simplicity–Parsimony Approach

**DOI:** 10.3390/ijerph191610117

**Published:** 2022-08-16

**Authors:** Emmanuel O. Amoo, Paul O. Adekola, Evaristus Adesina, Olujide A. Adekeye, Oluwakemi O. Onayemi, Marvellous A. Gberevbie

**Affiliations:** 1Demography and Social Statistics, College of Management and Social Sciences, Covenant University, Ota 112104, Nigeria; 2Department of Mass Communication, College of Management and Social Sciences, Covenant University, Ota 112104, Nigeria; 3Department of Psychology, College of Leadership and Development Studies, Covenant University, Ota 112104, Nigeria; 4Department of Business Management, College of Management and Social Sciences, Covenant University, Ota 112104, Nigeria

**Keywords:** young widow, in-laws, intimacy, widowhood rites, re-marry, sexual intimacy, health, widow rights

## Abstract

The incidence of short marital duration due to the demise of a husband that often exposes young widows to in-laws’ exploitation of the asset of the deceased spouses, without regard for negative health consequences and potential vulnerability to poverty has not been exhaustively investigated, especially in sub-Saharan Africa where 16% of adult women are widows. The study examined the coping mechanisms among the young widow (aged ≤ 40) who have experienced short conjugal relationships (≤5 years) and burdensome from in-laws. The research design followed a qualitative approach with the aid of semi-structured in-depth interviews among 13 young widows selected through snowballing and informant-led approaches in the purposively selected communities. Data collected were analysed using descriptive statistics and a thematic approach. The findings, among others, shows the median age of young widow as 29 years. All participants, except one, have faced exploitation from their in-laws over their husbands’ assets. All the participants desired to re-marry in order to: have a father figure for their children, have their own children or have more children. There is an absence of government support, but a few have received support from religious organisations. The author proposed attitudinal-change campaigns targeting the in-laws through accessible media and legislature that could challenge the exploitation of widows and unhealthy widowhood rites.

## 1. Background

Out of the basic demographic processes of life, birth and death occur to everyone, while marriage stands as the intersection between these two demographic polar axes. The event of marriage is universally unique, signifying the passage into adulthood and entrenched cultural responsibility, especially in the traditional cultural setting of sub-Saharan Africa. Birth is a desirable event in most families and is often welcome, but death is feared, not publicly wished, though sometimes it is accompanied by a celebration where the deceased is old in age or has attained riches maximally. Generally, birth is never mourned whether it occurs to the underage (boy or girl) or older individual, poor or rich, intact or separated family, or whether it happens in the city or countryside. However, death is always described as unwelcome and never prayed, for it is accompanied by considerable mourning irrespectively of the age involved. Death engenders some level of psychological bereavement, especially where it involves or is connected with young life, young marriage, or young parents. Conjugal bereavement is an unexpected marital disruption that could create psychological stressors and, improperly managed, spurs sickness (including mental health challenges) and death in the surviving spouse [1,2]. Whether the duration of the marriage is short or long, the common narratives over the experience of a widow are distress, ostracism, stigmatisation, and trauma [2,3,4]. 

Marriage and in-laws’ influences are conceptual intricacies in sub-Saharan Africa. Studies have confirmed the dominant role of in-laws’ behaviours in the determination of couple marital success across the globe and more dominant in sub-Saharan Africa [5,6]. The concept of marriage in Africa compels the wife to form familial bonds with the in-laws, a sort of engraftation into the groom’s family (which now becomes her new family). This makes the groom’s relatives become her new family members. In other words, the African marriage tenet compels the wife to leave her own family and becomes a family member of the groom [7,8]. Thus, in the case of anything happening to the wife, her husband or children, the needed consolation is expected to come from the groom’s relatives. It could therefore be disturbing and necessary for investigation where rather than the wife receiving such a gesture from the in-laws, the opposite is the case. Research, for decades, have established the societal broad challenges that widows face with respect to injustice, discrimination, harmful cultural rites [9,10,11,12,13] and that most of these challenges are culturally rooted. However, investigation of the specific challenges being inflicted by the in-laws, who are the supposedly immediate shield for the bereaved wife, is not conspicuous in the literature.

A report has highlighted that despite the principle of mutual respect and caring in a traditional African setting, the implementation is far from the truth when it comes to the treatment of widows and their rights [12]. Widows in Africa are faced with discrimination and unjust treatment [12]; in certain cultures, in Nigeria, widows are completely ostracised during the pre-burial period or subjected to dehumanising practices [3,10]. While the experience of widowhood, which thou transcends beyond one’s culture, has been documented in the literature, the experience of young widows and the dynamic of in-laws’ interference after the demise of the husband are not conspicuous in the literature. The study on dynamics of in-laws’ interferences in widow’s rights and how the widows with short marital experience are coping could highlight the needs of this special group of people and could be critical for social interventions.

In Indonesian society, widows (*janda mati*) suffer discrimination and disadvantage in everyday Indonesian social life. Their identity is often regarded with shame, they often experience economic hardship and feature more among the poor, and they could be open to salacious gossip or sexual harassment [14,15].

A report indicated that in Nepal, widowhood is highly stigmatized, and in most cases, the affected women always conceal their identity as widows as a shield against stigmatisation [16]. The report further indicated that while the concealment of identity is used as a vital coping mechanism, it also helps to manage the bereavement of their children [16].

There are quite a number of reports on the social consequences of widowhood, especially as it relates to health or health behaviour of older widows as peculiar to India, China, and Japan [17,18] (Aniruddha, 2013; Agrawal and Arokiasamy, 2009), including specific communities such as Vrindavan [9,19]. It has been reported that in Asian countries, widows are vulnerable to the torment of abusive tradition, indignities, discrimination including poverty [20]. Furthermore, a report in the United States indicated that the burden of widowhood is tilted towards older adults [21,22]. While in these climes, older widows are viewed as a burden by relatives, the situation surrounding the young widows is not conspicuous, especially in Nigeria. Lloyd-Sherlock, Corso and Minicuci (2015) reported deprivation among widows in five countries, namely: India, China, the Russian Federation, South Africa and Ghana, and specifically highlighted the existence of association between being a widow and having poorest wealth quintile in these countries [23]. While the study itself called for specific contextualisation of experience of widow as opposed to the generalisation that is commonly reported in many studies, further clarification by age segmentation and duration of the marriage could also add more knowledge to the body of literature.

A short marriage is a marital union duration that is ≤5 years [24]. In that type of marriage, both couples might relatively be inexperienced in marital issues, especially where both are in a similar or closer age bracket. The duration of the marriage is essential to intimacy, the shocks and the coping capability in terms of means of livelihood. The event of marriage has social connotations and predisposes the beginning of family formation that plays a significant role in society. The intertwining of the three phenomena of marriage, family and society, as well as their states, are therefore essential determinants of not only the marriage and the family itself but also fundamental to the health and wellbeing of the society and critical to sustainable economic development [4,25]. Marriage is expected to provide a long-life partnership between the couple and bring care and security for both wife and husband but especially serves as an economic refuge for the wife and children in sub-Saharan Africa, where the husband is expected to be the family head and the breadwinner [4,25]. However, the incursion of death could disrupt such expectations. It has been documented that the level of physical morbidities and death are higher among widows compared to their married counterparts [2,26].

When considering the higher life expectancy for women in the face of shorter longevity for men, conjugal loss may likely have a significant effect on women who are likely to live longer and cope with bereavement for longer years [27]. Where second marriage is allowed or even desired, the necessity for looking for another spouse after the demise of the first spouse with the associated stigma of widowhood may be more consequential than an ordinary loss of the spouse but remain a scar for life [28]. Where adequate comfort is not available, or there is no societal provision for the healing of this unexpected wound, neglect or inadequate care could engender more sorrow, regrets, and other health impacts, not only on the widow but on her children [14]. Globally, as of 2017, there were relatively 259 million widows with an average of two offspring each [29,30]. One in every 10 women of marital age is widowed, and one out of seven of them lives in abject poverty (Loomba Foundation, 2015). According to United Nations information on women, relatively 248 million out of the 350 million widowed people are women [31]. There has been a steady increase in the number of widows worldwide, with an increase of 9% between 2010 and 2015, which was ascribed to conflicts and diseases [29,30,32]. Country-wise, Indian and China are homes to the world’s largest proportion of widows, with an estimated 46 million and 44.6 million widows, respectively, in 2015 [29,30,33,34]. Among those in abject poverty, one in five lives in Afghanistan and Ukraine [29,30].

### Widowhood Rites

While the challenges faced by widows are diverse and not limited to a certain region, the intricacies surrounding widowhood in sub-Saran Africa call for concern. In sub-Saharan Africa, many widows are robbed of their spouse’s assets by the spouse’s immediate or extended family, as commonly practised in Angola, Bangladesh, Botswana, Ghana, India, Ivory Coast, Kenya, Malawi, Nigeria, Tanzania and Zimbabwe [29,30]. It is often not strange that numerous widows have been accused of witchcraft or as the source of the death of their spouses, and some are subjected to forced marriage, sexual cleansing or rituals and so on [29,35]. In certain countries in sub-Saharan Africa, the widow’s cleansing practice is a major rite where the widow is required to drink the water used to wash her dead husband’s body [29,30,36]. This cleansing could also include having sex with a relative of the deceased or a complete stranger [29,36] or being forced to marry any brother of the deceased [37]. These rites are performed outside the choice of the widow or any confirmation of the HIV status of either the woman or the person she is to have sex with. Other rites could include cutting of hair short, wearing dark clothes for certain days and remaining isolated at home [34] and, in some climes, isolated for almost 41 days.

The widow could be forced to drink the water used to wash the corpse of her husband because of the traditional belief that she will also die after drinking that water if she is responsible for the death of her husband. Several are brought before the family’s deity to swear their innocence [35]. Moreover, such widows could be vulnerable to stigmatisation and rejection because of the mere suspicion of having a hand in the death of their husbands. The stigmatisation could also be due to the superstitious belief that any associate friend of the widow (female or male) could also lose his/her own spouse. Holding the belief that the deceased was their son, a traditional tendency exists for the husband’s relations to dispossess the widow of her husband’s properties.

Irrespective of the cause of the death of the spouse, either suddenly or after a protracted illness, it could be devastating to experience an unplanned, sudden change in marital status, making the widow or widower go back to spinsterhood or bachelorhood with different lifestyles, intense emotions and anxieties about the future [38,39].

Before the abolition of *sati-practice* among the Hindus, the wife of the deceased husband will have to immolate herself either willingly or unwillingly (Loomba, 1993). *Sati* practice gained popularity in India with the belief that such suicide brings honour to the family and that the sins of both the husband and wife have been burnt off. Where such a wife refused to die, she is ostracised, regarded as ‘non-person’ who has dishonoured her family [11]. Such women are banned from attending religious functions and must have their heads shaved all the time [11]. Sati practice cuts across all ages and must be performed even if the wife is a five-year-old bride [11]. Notwithstanding the abolition of sati practice, today, widows are still facing hardship, marginalised and underprivileged widows and may often not be recognised in certain social structures [31]. While in Africa, especially in Nigeria, widows in the early mourning period must dress in black attire until the burial of their husband and for 40 days after. In Vrindavan (the historical city of widows), widows wrap themselves in colourful saris and white [19,40]. Vrindavan, a few distances from Delhi (India), regularly hosts hundreds of widows from across India, where they receive food and space for sleeping. Baugus (2019, p. 20) reported the experience of a widow in India as described by Delhi Psychologist Vasantha Patri as ‘physically alive but socially dead’ [9].

The challenges widows face are recognised across the globe to warrant the dedication of a special day—International Widows’ Day, mostly June 23 annually for acknowledging widows. It is a global observance day for spreading awareness of the plights of widows (such as poverty, injustice, and neglect) and encouraging the public and governments to come to widows’ aid across the world. The international was a craft of The Loomba Foundation but has been ratified by the United Nations since 2010 [13].

Life after the death of a spouse is not the same again, and such traumatic, devastating effects or eventual unpleasant emotions from the sudden bereavement would require adequate coping strategies in order to forestall negative psychological reactions, including illness and death. Coping mechanisms are generally described as behaviour, emotions, and thoughts that are used to manage the psychological stress of individuals in a manner that could improve their mental wellbeing and permits effective functioning. There are different types of coping mechanisms. These include problem-focused (or adaptive behavioural), emotion-focused, support-seeking, occupation-focused and avoidance coping [41,42,43,44]. While each type could be adopted, a simultaneous mixture of several types is also possible. In this context, the coping mechanisms are strategies embarked upon or used by the widows in adjusting to widowhood situations but which promote their wellbeing without further distraction in the day-to-day activities, notwithstanding the bereavement.

A number of studies have investigated the hidden injustices faced by widows [24,45], including the coping strategies that are being employed by the victims [2]. However, only a few studies have considered the age dimensions in widowhood [2], especially among those whose marriages last for a very short duration. Among the boggling questions is that, in traditional settings where cultural value systems of intact marriage are entrenched, what are their surviving ‘tricks’ among the young widows, especially in terms of stigmatism, traditional rituals, husband’s property, and so on.

The study used the simplicity–parsimony model, which has been defined as a combination of simplicity and parsimony models in analysing social behaviour with potential understanding among the target population irrespective of their educational attainment, especially in terms of statistical literacy [46,47]. The model of simplicity–parsimony is a derivative of the theories of parsimony and simplicity. The idea behind the simplicity model is to use available simplest methods to interpret observations on the fact that human minds could be more aroused towards situations that appear simple. The law of parsimony appraises the simpler solutions as the best problem-solving principle [46,48,49,50]. The model endorses the absence of complexity in the analysis and interpretation of phenomena. Specifically, the adopted simplicity–parsimony model is used to expatiate that using simple analysis could communicate to the general public irrespective of their educational attainment, especially in terms of statistical manipulations with high potential for achievement of desired change in the society.

Thus, following the patterns of Amoo et al. (2020), the importance of basic understanding of social phenomena could be indispensable for the effectiveness of interventions towards controlling or preventing social vices, risky behaviour or unhealthy traditional practices (such as widowhood rites), and including curbing the spread of diseases. Where the understanding of research output is lost or inadequate, abuse is inevitable. In most cases, the opportunity for comprehending statistical output could elude individuals who may be the most affected population due to their disadvantaged position in respect of statistical literacy [46,49,50]. Complex statistical results have the potential to widen the gap between the knowledge producers (researchers) and knowledge users, especially the policymakers [46,47]. The study, therefore, presents a descriptive analysis of narration from the young single widow and the coping strategies being employed using percentage distributions and thematic analysis without statistical complexity. The study is poised to find out the behavioural approach of the in-laws and the cooping experiences among the young widows. It asked questions on the reasons for delaying in re-marriage and/or the decision not to re-marry again. In this context, the term un-re-married is used to describe any young woman who has remained unmarried since the death of her husband.

## 2. Methods and Materials

### 2.1. Research Design

The study adopted a phenomenological qualitative approach aided by in-depth interview among the identified young un-re-married widow (aged < 40) who reside in area where traditional marriage systems are sacrosanct. The participants were drawn from potential widows who accepted to share their experiences. The eligible widows are those who have lost their spouse for ≤5 years and have remained unmarried till the time of the survey.

### 2.2. Research Instrument and Data Collection

The study used in-depth interview guide (patterned as a semi-structured interview) as the principal instrument in the data collection. After the initial development of the interview guide by the principal investigator, all members of the team reviewed the guide. Since members of the team who are also experts in qualitative research also served as interviewers, there was no formal training organised. However, several interview practices were held among the team members. Moreover, a pilot exercise was carried out with two widows outside the survey’s location. Responses were recorded with hand-phones. The outcome was used to refine and modify the initial interview guide. English language was used in the conduct of the interviews though this was occasionally intermediated or mixed with the use of local Pidgin English.

The study examined the challenges faced and coping mechanisms among young un-re-married widows. The interview asked questions about the duration of the marriage, the share of family responsibilities while spouse was alive, causes of the death of the spouse, and the family’s (wife’s and husband’s family) notion of her re-marriage. The respondents were also asked questions about their personal opinion on re-marriage, and reason(s) for delaying re-marriage until the current interview month or beyond. Question was asked on the intention not to re-marry. The interview was duly guided by the pre-developed tailored topics, and each of the interviews lasted only 50 min or less.

### 2.3. Data Analysis

Data gathered were analysed using systematic-content analysis [51,52,53]. First, the data collected were subjected to quality proofreading. All recordings performed during the interview were listened to a couple of times and transcribed accordingly. The transcribed responses were later matched with notes taken from the field. These processes provided opportunity for quality data and avoidance of duplication. Emerging concepts or major statements were coded and organised into categories. These categories formed the themes after thorough review and agreement among the authors on the concepts, major phrases or statements. The data were analysed manually, and this was made easy considering the limited number of respondents. In addition, the duration of each interview (≤50 min each) was manageable. A descriptive narrative approach was employed in the thematic organisation and analysis of the data. The authors employed Microsoft Excel and Microsoft Word in most of the organisation of the data. A Microsoft Word document was created detailing responses to each question. These were transferred into MS Excel, where cross-comparison was performed by age group: widow aged ≤ 34 years and widow aged > 34 years.

The results were benchmarked with existing literature. The data analysis was believed to adhere to qualitative research review guidelines (RATS) that place emphasis on the relevance of research questions, appropriateness of methods, transparency, and soundness of interpretive approach [51,52,53]. The author also followed the consolidated criteria for reporting qualitative research (COREQ) in the presentation of this research [51,52,53].

Finally, in observance of the ethical practices and the consent promises of data security, the records were deleted after the transcription.

### 2.4. Ethical Considerations

The process that led to the approval of the study comprised the submission of the study proposal to Covenant University Health Research Ethics Committee (CHREC) for review and the successful completion of the West Africa Bioethics Program and the Nigerian National Code for Health Research Ethics (Record ID: 43631126). Participant informed consent (Ref: SH/P-NIG 014/2021_08) for voluntary participation were directly obtained from each of the prospective participants who were selected through snowballing and informant-led approaches in the purposively selected communities. Major screening question is about young widows (aged ≤ 40) who have not re-married. The participants excluded those women whose marriages lasted more than 5 years. In addition, individuals who showed unwillingness to participate were excused from the interview. We also sought and obtained permission from identified senior persons where the prospective participant is living with relatives in the household.

Prior to the selection of the study participant, a preliminary community reconnaissance took place a few days before the main fieldwork. Verbal permissions were sought from heads of various communities selected. Each participant was duly informed that their participation is voluntary, that there is no material compensation for participating and that she may wish to discontinue at any stage of this interview, especially due to other schedules or whenever she feels uncomfortable with the interview. In addition to the above procedures, all respondents were again requested to re-affirm their voluntary interest to participate in the study on the first page of the questionnaire in a Yes or No response (Appendix: Ref: SH/P-NIG 014_A/2022_01). The research guide did not contain any identifier (such as names, addresses, e-mails or phone numbers). We also assured the participants of the confidentiality of their responses. Moreover, we limited the interview to only required information.

## 3. Results

### 3.1. Demographic Profile of the Participants

The widow shared their experiences during and after the loss of their spouse. Thematic analysis intermediated with content analysis was used in analysing the qualitative data collated from the in-depth interviews. The participants consisted of 13 young women aged 40 years and below. The age distribution revealed that relatively more than half (53.8%) were in the age group of ≤35 years, while 46.2% belonged to the 36–40 years’ age group. The median age of the participants is 35 years, signalling they are in their prime age of life. The median age at marriage is 29 years, and 15.4%, 46.2%, and 38.5% married at the age of 20–24, 25–29, and 30–34 years, respectively (Table 1). The educational attainment statistics show that all participants are educated, and their educational statuses cut across primary, secondary, and tertiary institutions. Those who have only primary education were 23.1%, the proportion of participants with secondary education is 46.2%, while those who have tertiary education are 30.8% (See Table 1). Over 60% of the participants were working at the time of the interview. The occupation categories varied from poultry/crop farming (23.1%), trading (such as buying and selling), and teaching 23.1%. The proportion of participants who are artisans and office clerical assistants is 15.4% each (Table 1).

Questions were asked about the duration of the marriage before the demise of their spouses. Responses received show that nine participants (69.2%) have been married for up to 5 years and below before the demise of their spouses, while only four participants (30.8%) have been married for 10 years and above. While four participants have not given birth (zero parity), six participants have only one child each, and 23.1% have between two and three children (Table 1).

### 3.2. Level of Intimacy Experienced with Late Husband

Discussions were had over the kinds and level of intimacy the participants experienced with their late spouses. The essence is to understand the level of bonding that was between the former couple and probe into how the gap has been filled since the death of the spouse. On the one hand, the specific questions were on sexual bonding, working or financial cooperation on intimacy between the couple. On the other hand, a probe was also made into crisis relationships, conflict, and the level of disunity between them while the husband was alive.

The guide is mediated with a quantitative measure of the level of intimacy using a six-level indicator, namely: (1) Sharing passion/pleasuring (sexual intimacy), (2) Sharing financial responsibilities (fiancé intimacy), (3) Relating experiences of fun/play (recreational Intimacy), (4) Share common tasks/working together (Work Intimacy), (5) Facing crisis and struggling (crisis relationship), and (6) Fighting and abusive (conflict intimacy). The descriptive information from the in-depth interview revealed that every participant had experienced sexual intimacy with the late spouse. However, the level in other intimacy indicators differs. In terms of financial responsibility, relatively half of the total respondents shared financial responsibility with their spouses; a total of 23.1% either handled the financial upkeep of the family alone or made the sole responsibility of the husband and not both (Table 2). More than 50% of the respondents enjoyed recreation intimacy with their spouses, and 15.4% never (Table 2). While less than half (precisely, 46.2%) worked together, 30.8% never worked together or shared household chores together. A few of the couples experienced a crisis regularly before the demise of the husband, and almost half, 46.1%, do so not often (Table 2). In addition, the result revealed that 15.4% of the total respondents fought with their spouse very often before the latter died, as presented in Table 2.

Responses from the interview show that an overwhelming proportion (11 participants out of 13, i.e., 84.6%) indicated that they had sexual intimacy with their spouses very often. However, two participants indicated otherwise. Excerpts from the interview are shown below:


*“Sexual bonding is a normal thing; of course, we had sexual intimacy throughout his life. We do it as often as possible, beside the fact that we wanted to have children”.*
Widow, aged (≤34 years)

The variant response was from two participants who reported that the love bond between them and their spouse diminished gradually because of their childlessness:


*“Apart from the normal sexual relationship, the love was not there as expected. Every husband expects children from the marriage and the moment such seems delayed, his care or love began to diminish”.*
Widow, aged (≤34 years)

Relatively half of the participants expressed that they were emotionally attached to their spouses. A few excerpts indicated they had good feelings about themselves.


*“We do most things together, going for visitation, attending social events including going to church (or mosque) together. My man was a nice man, he mostly does nothing without my comment (input). He takes decision on his own but mostly inform me before the final execution of such agenda”.*
Widow aged (≥34 years)

### 3.3. Conflict, Crisis and Disunity

While the majority expressed that they shared ideas (e.g., new business information) with their spouses, a woman reported that she worked with her husband in the same establishment. However, another woman reported that she had been living separately before the demise of her husband. She affirmed negatively to most questions on intimacy (such as no joint decision taking, non-sharing of ideas, and no joint attendance in social events).

Three participants expressed divergent experiences in terms of emotiveness. The responses suggested that they might not be in mutual tune with their spouses and perhaps lacked emotional intimacy.


*“A woman responded: “It became obvious in less than 6 months of our marriage that our thoughts were mostly not in the same direction. We hardly agreed on anything, so I ‘left it for him’ (often allowed him to have his way).”*
Widow aged (≥34 years)

### 3.4. Family Relationship

One of the focuses of the study is to examine the average disposition of the in-laws towards the widows after the demise of their husbands. Information gathered indicated that the sub-Saharan African tradition of old where decease relative claims properties of the decease still subsist. Participants general responses signalled that husband’s property belongs to his parents and not his wife’s. Such property could be demanded or seized from any widow at the discretion of the deceased family. Such property could also be shared or transferred to another person or group depending on the consensual agreement in that family or the decision of the eldest in-law. How far this has been propagated or strengthened was confirmed by the testimonies of a few of the widows in this study.

A woman in her late 30s expressed that the challenges she faced in the marriage started before the demise of her husband. She complained about her in-laws’ interference even before the official consummation of the marriage. Excerpts from her responses read thus:


*“From the beginning, I knew the marriage might not last, though I was not thinking it is death that will separate us. Why, because, the family interference was just too much. The family perceived me as obstacle to their access to his (husband’s) wealth. Me, I was trying to make my husband free from being ‘teleguided’. The family always wanted him to be securing permission form them even over every matter. It doesn’t work that way”.*
Widow aged (≥35 years)

Another widow (aged < 35 years) explained that she lost her husband about three years when she was just 27 and had not had a child for the man. That notwithstanding their average lives, living in an uncompleted building, the in-laws’ eyes were on the building.


*“I eventually left the site (i.e., house) for them (in-laws) and never made any inquiry about it till date. He left with me an unsettled quarrel between him and his family. I was often the suspect as the brain behind his disposition to the family. It started with hostility that graduated to open confrontation. I had no option than to vacate the place and moved here (a different location). I have been abandoned totally by my in-law’s relatives”.*
Widow aged (≤34 years)


*“You know I was still hoping for a child. I became a suspect concerning his death. Although, I was not alleged openly, but the body language around was implying estrange relationship. I knew they were considering my sadness as pretense. The payback was to take away his belonging from me”.*
Widow aged (≥35 years)

### 3.5. Why Have You Not Re-Marry?

The issue of re-marriage was discussed. Realising how personal the issue is, the in-depth interview approach assured confidentiality and encouraged the participant to open up and freely discuss with the interviewer. We explored the intention to re-marry or not to re-marry and the reason(s) for delaying that intention.


*“I may not know how they (late husband family) will feel if I re-marry now. It’s just like three years now. I was actually not sent away by the family; they are nice people. We are still in touch today. I think I should give myself or them more time because of the love I had for their late son.”*
(Widow, aged (≤34 years)


*“It is not that I don’t want to marry. I’m only being careful and this time around, careful about the family members. It is essential I have a father figure for my son. Besides, I still want to have more children.”*
(Widow, aged (≤34 years)


*“My focus now is on my child. I am trying not let my child feel any father-gap. I cannot be thinking of re-marriage.”*
(Widow aged (≥34 years)


*“Yes, I’m planning now to re-marry but not without my children. Their desire was for me to release my two kids to them while I can leave the family, and start a new life. Of course, I needed to start a new life but not leaving my children to any stranger. My prospective husband must marry me and be able to take care for my two kids any other children that I am going to bear for him.”*
(Widow, aged (≥34 years)

### 3.6. Support Received

The study further investigated the frequency of support received by each of the participants either from relatives, religious organisations, employer or workplace colleagues, and the community association. Notwithstanding the smaller number of participants involved in this study, a pre-coded question was used to assess to rate the frequency of support received on a 4-Likert scale, namely: (1) Very often, (2) Often, (3) Not often and (4) Not at all. The response collated showed that there was relatively no general regular support for all the young widows interviewed except from family of the widows and their religious organisations (Table 3). Relatively, one out of every five widows has received support from her own family members and religious organisations. Information received from the participants indicated that there was no support received from the government (either from the national, state or local level). The general responses compiled are presented in Table 3.

### 3.7. Coping Mechanism

While losing a spouse could engender emotional experience, especially where the relationship is short-lived, the understanding of how they are surviving and managing a number of issues could be of immense foundation for teaching, encouraging and counselling others in order to relieve them of their pains, depression due to the demise of their spouses. The study conceptualized coping as the behaviour or steps taken by the young widow to overcome or manage the in-laws’ disposition toward her after the demise of her husband (who was also the in-laws’ son or relative). It considered the economic difficulties encountered due to sudden change in marital status (arising from the death of the husband). Precisely, it explored how the widows are coping with in-laws’ requests or attempts to dispossess the widow of her husband’s estate or other belongings.

Specific questions asked on how the participants have been coping brought to fore information on the survival and management strategies in the areas of finance, sexual life, relationship with in-laws, religious activity, and so on. A few coping strategies reflected in the discussion are: reduction or stop in terms of worries; the majority would not want to be regarded as a widow but as a single woman; a few respondents expressed a loss of sexual drive; having sexual partnership which (according to one of them) could translate to marriage. In addition, a few are keeping distance from the in-laws (a king of avoidance, especially where they belong to the same religious assembly, about two widows have resigned to fate.


*“Raising this child alone is dreadful, I have changed jobs like three times within 3 years now. I just have to fend for my child for now.”*
Widow aged (≥34 years)


*“I have experienced two lives now: first as single, as married and now as ever-married but single again. Notwithstanding, I need to start an entire new life now. I must surely be remarried, when, is what I can’t tell.”*
Widows, aged (≤34 years)


*“I have changed my accommodation, now leaving far away from that community.”*
Widow, aged (≤34 years)


*“I have stopped attending our church, I don’t want to be meeting them again, I keep a distance. Once a while, I do call anyone of them but meeting them often would be reminding me of the past.”*
Widows aged (≥ 34 years)


*“Are you saying I should not move on? I have to move on, planning to re-marry means I am already dating again. Yes, I have ‘male’ friend now.”*
Widow, aged (≤34 years)

## 4. Discussion

The study examined the coping mechanisms among young widows (aged ≤ 40) who have experienced short conjugal relationships (≤5 years) and faced burdensome from the in-laws. This is quite different from other studies that have either considered bereavement [1,2,28,54,55] or coping strategies among widows irrespective of their age classifications or studies conducted without consideration of marital duration [3,4,34]. The study of widows’ experiences in the midst of in-laws influences and the sacrosanct marital system in Nigeria could be crucial for legislation and re-examination or enforcement of existing laws for women and girls’ rights and the abolition of harmful traditional rites that could affect health and diminishes the dignity of womankind.

The study dealt with the issue of importance for the quality of life that is related to seemingly unnoticed young groups of women with disrupted marital relationships by circumstances of husband death and exploitation by supposedly relatives (the in-laws). The problem addressed is important and interesting, but unfortunately, the prevalence is in many environments, not only in Africa but across other continents. It highlighted the limited or narrowing of the system of support for widows in the place of study.

Specifically, the study, by implication, is an awesome revelation of underlying threats to the achievement of promotion of the gender equality agenda (SDG_5) in a system where rights to properties are restricted by undue interference from the in-laws. It depicts an irony (of efforts) in the drive to reduce poverty (SDG-1) with no conspicuous government support for widows, especially the young ones. The challenge, if unresolved, could hinder the achievement of the UN goal of elimination of all forms of discrimination against women or the clamour for the support of widows or empowerment of widows in Asia, Africa and Latin America, especially as being championed by the UN Women advocates [19]. The study has signalled the mirage in achieving holistic, healthy lives and promotion of wellbeing across all ages (SDG_3) where certain categories of the population could be made to face traditional-system-infected challenges where widows are denied the rights to their husband properties. Where such practice goes on unsuccessfully challenged, it could reinforce the circle of poverty [37]. The study is also an approach to impress upon the intertwined three phenomena: (1) the marriage, (2) family, and (3) the society that are indispensable in the achievement and maintenance of sustainable development in health and wellbeing [25].

While the onus of family livelihood unexpectedly rests on the widow, for the survival of the children (where available) or self, notwithstanding her bereavement period, trying to come to the reality of the death of her spouse, the widow might need to contend with husband’s relations who could be ready to retrieve the deceased’s assets/properties from the widow. These conditions could be inimical to a healthy life and current and future economic productivity. Another distinguishing feature of this study is the conceptualization of widow as distinct from the known description where ever-married women who have lost their husbands and those who were not formally married (but lost their partners) were grouped together [18,55]. In this study, a young widow is captured as the only ever-married woman (≤40 years) whose husband died in the last 5 years and has remained un-re-married until the survey period. The study is an extension of other studies that have highlighted concerns for families where the expected long-life partnership between a couple is disrupted by the demise of the supposed breadwinner of the family [2,4,26].

The study is another highlight of short life expectancies among men and the potential length of years the widow has to endure coping alone if she refuses to marry again, either intentionally or by other circumstances. The findings support other studies that have highlighted that the widow may have to be coping with bereavement for longer years, stigmatisation, and family support stringencies, especially where second marriage is proven difficult [27,28,54]. It emphasised the essence of intimacy lost between young couples due to the demise of the spouse and a way forward for such young widows even where widowhood support is not available.

The study highlighted the burdensome imposed on the widows by the in-laws, especially where the marriage is childless. It signalled the continued significance of procreation in the traditional African marriage system, with covert and overt stigmatisation trailing any wife where the couple failed to have at least a child [2]. The aftermath of such stigmatisation cannot be benign to the health and wellbeing of the woman (especially those in childless status) and could definitely affect her economic productivity either at the homestead or in formal occupation [3,4,32]. Moreover, while certain widowhood rites, such as having sex with strange or deceased relatives, could engender contracting of infections, others, such as isolation for a certain number of days, could engender suicidal thoughts and spur health deterioration [34].

## 5. Limitations of the Study

The restriction of the study to only young widows (aged ≤ 40) narrowed the study coverage but enhanced the focused nature of the study with specific insights toward the targeted population for policy intervention. In addition, the exclusion of certain variables such as religious affiliation and ethnicity befogged the understanding of cultural intricacies in this study. Furthermore, the limited number of participants covered prevented extensive statistical analysis and further generalisation of the findings beyond the location of the study. However, the analytical presentation using both qualitative and quantitative approaches provided an adequate guide towards the policy recommendation from this study.

## 6. Conclusions

The study has provided an overview of the coping mechanisms among the widows in three distinctive scenarios of the young widow (aged ≤ 40) who experienced only shorter conjugal relationships (≤5 years) and faced exploitation from the in-laws. It concludes that the sudden loss of essential intimacy between a couple, followed by exploitation from in-laws and the presence of lack of regular support for the widows, are tripartite tragedies for any young widow. Therefore, considering the import of in-laws taking over the assets of the deceased (who invariably was their kin), without consideration of eventual deprivation of the wellbeing of their son’s wife (now the widow), the author proposed attitudinal-change campaigns targeting mainly the in-laws through accessible media, especially such media that can reach every traditional setting. Legislature challenging the exploitation of widows by the in-laws’ is exigent in order to protect their rights and stop unhealthy widow rites. In addition, the number of widow participants in this study indirectly signalled the frequency of death among young men in the locations of the study, which should also call for concern among the stakeholders.

## Figures and Tables

**Table 1 ijerph-19-10117-t001:** Demographic profile of the respondents.

Age Group	Number	Percent	Age Group at Marriage	Number	Percent
≤34 years	7	53.8	20–24 years	2	15.4
35–40 years	6	46.2	25–29 years	6	46.2
**Total**	**13**	**100.0**	30–34 years	5	38.5
**Mean = 36, median = 35**			**Total**	**13**	**100.0**
**Educational attainment**			**Median = 29**		
Primary Education	3	23.1	**Current Occupation**		
Secondary Education	6	46.2	Farming (Poultry, cropping)	3	23.1
Tertiary Education	4	30.8	Trading (buy and selling)	3	23.1
**Total**	**13**	**100.0**	Artisans	2	15.4
**Duration of marriage**			Office clericals	2	15.4
≤5 years	9	69.2	Teaching	3	23.1
6–10 years	4	30.8	**Total**	**13**	**100.0**
**Total**	**13**	**100.0**	**Children ever born**		
**Working status**			Zero parity	4	30.8
Yes	8	61.5	One child	6	46.2
No	5	38.5	2–3 children	3	23.1
**Total**	**13**	**100.0**	**Total**	**13**	**100.0**

**Table 2 ijerph-19-10117-t002:** Level of intimacy before the demise of the husband.

Intimacy Indicators	Very Often	Not Often	Not at All	Total
Sharing passion, pleasuring (sexual intimacy)	13 (100)	-	-	13 (100)
Share financial responsibilities (fiancé intimacy)	7 (53.8)	3 (23.1)	3 (23.1)	13 (100)
Relating experiences of fun/play (recreational Intimacy)	9 (69.2)	2 (15.4)	2 (15.4)	13 (100)
Share common tasks/working together including home chores (Work Intimacy)	6 (46.2)	3 (23.0)	4 (30.8)	13 (100)
Facing crisis and struggling (crisis relationship)	3 (23.1)	6 (46.1)	4 (30.8)	13 (100)
Fight and abusive (conflict intimacy)	2 (15.4)	4 (30.8)	7 (53.8)	13 (100)

**Table 3 ijerph-19-10117-t003:** Frequency of support received.

Nature of Support Received	Often	Not Often	Not at All	Total
Support from spouse relatives	-	2 (22.2)	7 (77.8)	9 (100)
Support from your own family relatives	3 (23.1)	5 (38.5)	5 (38.5)	13 (100)
Support from religion organisations	3 (23.1)	10 (76.9)	-	13 (100)
Support from government (local/state/national)	-	1 (7.7)	12 (92.3)	13 (100)
Support from community members/associations	-	10 (76.9)	3 (23.1)	13 (100)
Support from work place	-	4 (30.8)	9 (69.2)	13 (100)
Support from Counsellor(s), Psychologist(s)	-	7 (53.8)	6 (46.2)	13 (100)

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
