# Peer review of "Young Single Widow, Dynamics of In-Laws Interference and Coping Mechanisms: Simplicity–Parsimony Approach"

_ijerph, 2022, doi:10.3390/ijerph191610117_

Round 1
Reviewer 1 Report
The topic is great. It captures real social phenomena, however, the topic is quite too generic. The novelty must be emphasized more.
Some questions and suggestions regarding the essay:
- After this claim, it is better to show/cite some studies:
Whether the duration of the marriage is short or long, the common narratives 48 in the experience of widow are distress, ostracism, stigmatisation, and trauma (Ugwu et 49 al., 2020).
For instance, what studies of stigma talk about this.
Mechanical issues:
- The study on bereavement cannot not
- (Ugwu et al., 2020)o,
- What is the difference between widow’s life in sub-Sahara and other countries? Is there any specific uniqueness? Providing comparative studies between sub-Sahara and other countries is recommended
- Stories of widow Sati explored by Ania Loomba could enrich the analysis.
- Please mention if the ethic was issued by an external party.
- “9 participants (69.2%) have been married for up 223 to 9 years and below” – it would be better to mention the minimum year the participants remarried.
- The introduction sounds great and promises a unique topic; unfortunately, the result sounds generic. It would be better to focus on something unique and interesting. The essay might be more interesting if it focuses more on the dynamic of family interference
Author Response
Response to the Reviewers’ Comments on Manuscript entitled “Young Single Widow, In-Laws Burdensome and Coping Mechanisms: Simplicity-Parsimony Approach”, ijerph-1689312.
General:
We appreciate the reviewers comment received. We have dutifully addressed all the omission/oversights and followed the advice and suggestions from all the reviewers.
- Topic
The topic was amended to reflect the dynamic of in-laws interference.
Current title: Young Single Widow, Dynamics of In-Laws Interference and Coping Mechanisms: Simplicity-Parsimony Approach
- More literature has been added (line as advised) Line 52 (Old page 48)
- The Sati practice among the Hindus
- We appreciate the Reviewer for directing our attention to the Sati practice among the Hindus. This has added beauty to the story presented in this article.
- Experience in other countries and culture has been introduced. We cite experience of widows in Nepal and Indonesia.
ewed).
- Grammar Check/Editing
A consideration grammar proofreading has been conducted on the manuscript.
- Ethical consideration section has been provided
- References
We have a few more references
Amoo, E. O., Omideyi, A. K., Fadayomi, T. O., Ajayi, M. P., Oni, G. A., & Idowu, A. E. (2017). Male reproductive health challenges: appraisal of wives coping strategies. Reproductive health, 14(1), 1-10. https://link.springer.com/content/pdf/10.1186/s12978-017-0341-2.pdf
Clark J. (2003). RATS guidelines. How to peer review a qualitative manuscript. In Peer Review Health Sciences, 2nd Edition (eds.) Godlee F. and Jefferson T. BMJ Books, London. 2003. p. 219-235. https://bmjopen.bmj.com/content/suppl/2012/01/12/bmjopen-2011-000138.DC1/BMJ_Open_IMG_Physician_Migration_RATS_Checklist.pdf
Dube, M. (2022). When the Challenges of Widowhood Extend to Childcare: Essential Considerations for Social Work Practice. Social Sciences, 11(5), 225. https://doi.org/10.3390/socsci11050225 https://www.mdpi.com/2076-0760/11/5/225/htm
Loomba, A. (1993, October). Dead women tell no tales: Issues of female subjectivity, subaltern agency and tradition in colonial and post-colonial writings on widow immolation in India. In History workshop (No. 36, pp. 209-227). Oxford University Press. https://www.jstor.org/stable/4289259
Parker, L., Riyani, I., & Nolan, B. (2016). The stigmatisation of widows and divorcees (janda) in Indonesia, and the possibilities for agency. Indonesia and the Malay World, 44(128), 27-46. https://doi.org/10.1080/13639811.2015.1111647
Tong, A., Sainsbury, P., & Craig, J. (2007). Consolidated criteria for reporting qualitative research (COREQ): a 32-item checklist for interviews and focus groups. International journal for quality in health care, 19(6), 349-357. https://doi.org/10.1093/intqhc/mzm042

Reviewer 2 Report
The text may be of interest to a reader from a different region of the world, or a culture different from that in the African region. In this sense, it has cognitive value. The authors describe the unfavorable situation of young widows as a result of legal regulations concerning the financial situation of widows in Nigeria. The authors pointed to the need to take actions to change social awareness and overcome stereotypes present in the culture of this region of the world. This goal is undoubtedly right, although its implementation requires the support of state institutions, the media, changes in educational programs, and thought patterns - and requires a long time. The threats to the mental and somatic health of women and their children were highlighted, as well as to the physical health of possible new partners that are related to rituals (e.g. forced sex).
I do not reject the supposition that a linguistic correctness check could improve the quality of the text.
The authors wrote that one of the objectives of the reported study was as follows: “The study aims to understand the behavior of the in-laws and the coping experiences of young widows. The reasons for delays in re-marrying and / or the decision not to remarry have been investigated.
I did not find information in the text about coping strategies or reasons for postponing the decision to remarry.
The applied research approach allows for the collection of material that, even if subjected to a simple statistical analysis, would make it possible to obtain an answer to the question about coping by "young widows". However, it is difficult to recognize how the authors understand "coping" or what is the topic of coping? Is it about dealing with the negative emotions triggered by your husband's death? How do you mourn the loss of your husband? Is it about dealing with economic difficulties, with a change in social status (married woman - a single widow) as a consequence of her husband's death, including dealing with "rituals" (e.g. drinking water that was used on the body of the deceased, or forcing intercourse with a family member of the deceased husband, etc.) and the traumatic experiences they caused? Is it about dealing with family requests for a widow's estate (goods left by the deceased)? It is not easy to find the answer to these questions - in the text. The reasons for the refusal to remarry have not been clarified, but it has been noted that remarriage may improve the situation of the widow and her children.
I do not understand the idea expressed in the sentence: “Additionally, the concern among stakeholders should be an indirect emphasis on the frequency of deaths of young men in the locations studied. "
This thought has been expressed at least twice.
I suggest to consider the possibility of working through the text in such a way as to 1) define the mechanisms of coping; 2) measure their frequency; 3) recognize the factors determining their differentiation. It is possible without the use of complex analyzes - the more so as the study group was small. Data obtained from conversations with widows can be the basis for such analysis.
(The use of narrative is also possible, although it may prove more difficult.)
The text may be of interest to a reader from a different region of the world, or a culture different from that in the African region. In this sense, it has cognitive value. The authors describe the unfavorable situation of young widows as a result of legal regulations concerning the financial situation of widows in Nigeria. The authors pointed to the need to take actions to change social awareness and overcome stereotypes present in the culture of this region of the world. This goal is undoubtedly right, although its implementation requires the support of state institutions, the media, changes in educational programs, and thought patterns - and requires a long time. The threats to the mental and somatic health of women and their children were highlighted, as well as to the physical health of possible new partners that are related to rituals (e.g. forced sex).
I do not reject the supposition that a linguistic correctness check could improve the quality of the text.
The authors wrote that one of the objectives of the reported study was as follows: “The study aims to understand the behavior of the in-laws and the coping experiences of young widows. The reasons for delays in re-marrying and / or the decision not to remarry have been investigated.
I did not find information in the text about coping strategies or reasons for postponing the decision to remarry.
The applied research approach allows for the collection of material that, even if subjected to a simple statistical analysis, would make it possible to obtain an answer to the question about coping by "young widows". However, it is difficult to recognize how the authors understand "coping" or what is the topic of coping? Is it about dealing with the negative emotions triggered by your husband's death? How do you mourn the loss of your husband? Is it about dealing with economic difficulties, with a change in social status (married woman - a single widow) as a consequence of her husband's death, including dealing with "rituals" (e.g. drinking water that was used on the body of the deceased, or forcing intercourse with a family member of the deceased husband, etc.) and the traumatic experiences they caused? Is it about dealing with family requests for a widow's estate (goods left by the deceased)? It is not easy to find the answer to these questions - in the text. The reasons for the refusal to remarry have not been clarified, but it has been noted that remarriage may improve the situation of the widow and her children.
I do not understand the idea expressed in the sentence: “Additionally, the concern among stakeholders should be an indirect emphasis on the frequency of deaths of young men in the locations studied. "
This thought has been expressed at least twice.
Proposes to consider the possibility of working through the text in such a way as to 1) define the mechanisms of coping; 2) measure their frequency; 3) recognize the factors determining their differentiation. It is possible without the use of complex analyzes - the more so as the study group was small. Data obtained from conversations with widows can be the basis for such analysis.
(The use of narrative is also possible, although it may prove more difficult.)
Author Response
Response to the Reviewers’ Comments on Manuscript entitled “Young Single Widow, In-Laws Burdensome and Coping Mechanisms: Simplicity-Parsimony Approach”, ijerph-1689312.
General:
We appreciate the reviewers comment received. We have dutifully addressed all the omission/oversights and followed the advice and suggestions from all the reviewers.
- Topic
The topic was amended to reflect the dynamic of in-laws interference.
Current title: Young Single Widow, Dynamics of In-Laws Interference and Coping Mechanisms: Simplicity-Parsimony Approach
- More literature has been added (line as advised) Line 52 (Old page 48)
- Coping mechanisms
A whole paragraph been devoted to Coping Mechanisms in terms of concept, types and contextual relevance to this study. We mentioned the different types (e.g. problem-focused, emotion-focused, etc) and supported this with relevant literature e.g. (Billings & Moos, 1981; Folkman & Tedlie, 2004; Lazarus & Folkman, 1984; Weiten & Lloyd, 2008)
- We also have a separate result section for Coping mechanisms as being employed by the participants.
- Data Analysis procedure: The section has been broadening and more detailed that the previous. Specific insertion are as follows:
Data gathered were analysed using systematic-content analysis (Amoo, et al, 2017; Clark, 2003; Tong, Sainsbury & Craig, 2007). First, the data collected were subjected to quality proof reading. All recording done during the interview were listened to a couple of times and transcribed accordingly. The responses were later transcribed and matched with notes taken from the field. These processes provided opportunity for quality data and avoidance of duplication. Emerging concepts or major statement were coded and organised into categories. These categories formed the themes after thorough reviewed and agreement among the authors on the concepts, major phrase or statements. The data were analysed manually and this was made easy considering the limited number of respondents. In addition, the duration of each interviews (≤ 50 minutes each) was manageable. A descriptive narrative approach was employed in the thematic organization and analysis of the data. The authors employed Microsoft Excel and Microsoft Word in most of the organisation of the data A Microsoft Word document was created detailing responses to each question. These were transferred into MS Excel where cross- comparison was done by age group: Widow aged ≤ 34 years and widow aged > 34 years.
The results were benchmarked with existing literature. The data analysis was believed to adhered to qualitative research review guidelines (RATS) that places emphasis on the relevance of research questions, appropriateness of methods, transparency, and soundness of interpretive approach (Amoo, et al, 2017; Clark, 2003; Tong, Sainsbury & Craig, 2007). The consolidated criteria for reporting qualitative research (COREQ) was used in presentation of this research (Amoo, et al, 2017; Clark, 2003; Tong, Sainsbury & Craig, 2007).
Finally, in observant of the ethical practices and the consent promises of data security, the records were deleted after the transcription.
- Result section
A separate section has been added that contain identified coping mechanisms among the respondents (widows interviewed).
- Grammar Check/Editing
A consideration grammar proofreading has been conducted on the manuscript.
- References
We have a few more references
Amoo, E. O., Omideyi, A. K., Fadayomi, T. O., Ajayi, M. P., Oni, G. A., & Idowu, A. E. (2017). Male reproductive health challenges: appraisal of wives coping strategies. Reproductive health, 14(1), 1-10. https://link.springer.com/content/pdf/10.1186/s12978-017-0341-2.pdf
Clark J. (2003). RATS guidelines. How to peer review a qualitative manuscript. In Peer Review Health Sciences, 2nd Edition (eds.) Godlee F. and Jefferson T. BMJ Books, London. 2003. p. 219-235. https://bmjopen.bmj.com/content/suppl/2012/01/12/bmjopen-2011-000138.DC1/BMJ_Open_IMG_Physician_Migration_RATS_Checklist.pdf
Dube, M. (2022). When the Challenges of Widowhood Extend to Childcare: Essential Considerations for Social Work Practice. Social Sciences, 11(5), 225. https://doi.org/10.3390/socsci11050225 https://www.mdpi.com/2076-0760/11/5/225/htm
Loomba, A. (1993, October). Dead women tell no tales: Issues of female subjectivity, subaltern agency and tradition in colonial and post-colonial writings on widow immolation in India. In History workshop (No. 36, pp. 209-227). Oxford University Press. https://www.jstor.org/stable/4289259
Parker, L., Riyani, I., & Nolan, B. (2016). The stigmatisation of widows and divorcees (janda) in Indonesia, and the possibilities for agency. Indonesia and the Malay World, 44(128), 27-46. https://doi.org/10.1080/13639811.2015.1111647
Tong, A., Sainsbury, P., & Craig, J. (2007). Consolidated criteria for reporting qualitative research (COREQ): a 32-item checklist for interviews and focus groups. International journal for quality in health care, 19(6), 349-357. https://doi.org/10.1093/intqhc/mzm042

Round 2
Reviewer 1 Report
I can see hard effort from authors in editing the manuscript and adding some more information. Authors have also added similar issues in other countries, however, the authors can still add more in the introduction what is the difference between this research and prior studies. Taking examples from Middle East countries should be considered.
Also, please also double-check the connection words between the old paragraph and the additional ones.
Author Response
Reviewer 1
Query: Further clarification between this research and prior studies taking examples from Middle East
We have added various line of story as reported in the respect of the same subject in Middle East, citing the vulnerability of widow in India, China, Japan, including Vrindavan community, a suburb in India. While these studies treated the issues related to widow in general, the population segment we addressed (young widow with little marital relationship experience have not been contextualise or conspicuous in the literature, more especially, as it relates to Nigeria context where in-laws domineering power is sacrosanct.
Query: Double-check the connection words between the old paragraph and the additional ones
Response: This has been adequately perfected. About one or two paragraphs have been combined or moved to join relevant paragraphs /pages.

Reviewer 2 Report
The article deals with the issue of importance for the quality of life of individuals and communities of different generations.
The problem is important, interesting and (unfortunately) present in many environments - not only in Africa or Asia - due to the limited / narrowing of the system of support for widows and children (orphans).
The question is whether the title reflects the content of the article. I wonder if I understand the title of the article correctly.
I notice certain limitations in the way of presenting the collected material.
On the basis of the title of the article, more information can be expected about the in-laws' interference in the life of a young widow. On the other hand, the content contains information on 1 / the in-laws' interference during the marriage and 2 / on the behavior of the parents-in-law / family of the deceased husband - after his death. This enables reflection on the possible change in the behavior of the in-laws over time - as it was during the husband's lifetime - as it is after his death. The collected matter enables this, at least to some extent.
It contains interesting information about the relationship with the husband, including their intimacy - this applies to the period of marriage. Information is also provided about the woman's feelings, satisfaction or disappointment with her husband.
Do the authors have data to identify differences / changes in a woman's situation after her husband's death, depending on the quality of her relationship with her husband during the marriage? Including his family's behavior, her expectations of her widow?
Among the statements of the surveyed women about the relationship with the deceased husband's family - there are also positive ones.
Not much information has been provided on coping mechanisms - in my opinion they are not very clear-cut, rather general. Is it not possible to try to compile a table - to distinguish between escape strategies, task strategies, and emotion-focused strategies? (the authors wrote about it in the text) - and enter / assign the statements of the surveyed women to these categories / types of strategies? It was mentioned in the text that some of the surveyed women are working - can't paid work be considered a way of coping? If so, what kind? Escape or task? It should be clarified - what is a woman doing with the wages from this job? Does he give away earnings / income from her own work to the family of the deceased husband or spend it on satisfying her-self and the child's needs?
A certain vagueness / cautiousness of expression - perhaps reflecting the influence of culture - concerns the process of socialization of girls / women in a given culture. Please note the possibility of introducing three selection criteria to the test group - instead of three scenarios.
Please consider, among others over the sentence: The study contextualised coping as how the young widows are dealing with situation that arises as a result of interference the in-laws after the demise of the husband (who was also their late son).
(If the statements were recorded, maybe it is possible, for example, to identify emotions in the surveyed women? This leads to a slightly different way of collecting data - the inclusion of observational data. This is rather a proposition for future research: going beyond the narrative - and content analysis.)
Author Response
Reviewer II
General Response
We appreciate the reviewers for their in-depth analysis and thorough screening of this report. We have attempted to provide answers to all the queries. However, any query that could not be answered within the data collected (or beyond our analytical capacity) has been treated as part of the limitations of this study. However, we provide highlight on the few one we could answer without diverting our attention to a new topic entirely.
Query: Do the authors have data to identify differences / changes in a woman's situation after her husband's death, depending on the quality of her relationship with her husband during the marriage? Including his family's behaviour, her expectations of her widow?
Response: No. Our attention was just on the young widow experience with their in-laws and how they have overcome such challenges posed by the in-laws. It should be noted that: the women surveyed are young and the death of their husbands is still very fresh (i.e. very recent), any attempt to ‘dig deeper’ in the interview could heighten other emotional issues which we may not have the capacity to control.
Query: On the basis of the title of the article, more information can be expected about the in-laws' interference in the life of a young widow. On the other hand, the content contains information on 1 / the in-laws' interference during the marriage and 2 / on the behavior of the parents-in-law / family of the deceased husband - after his death. This enables reflection on the possible change in the behavior of the in-laws over time - as it was during the husband's lifetime - as it is after his death. The collected matter enables this, at least to some extent.
Response: We only focused on the disposition of in-laws to the widows, especially in terms of husband properties. It is based on the culture of parents’ claiming ownership of their son’s asset/property, disregarding the ‘existence’ of his wife or even the son’s children. The culture is so much endemic that the wife could be pushed out of her husband house (or apartment) because ‘whatever the son owns, belongs to his parent’. That is what we have addressed – “in-law interference”.
Query: It was mentioned in the text that some of the surveyed women are working - can't paid work be considered a way of coping? If so, what kind? Escape or task? It should be clarified - what is a woman doing with the wages from this job? Does he give away earnings / income from her own work to the family of the deceased husband or spend it on satisfying her-self and the child's needs?
Response: Considering the political situation in the country (Nigeria), the issue of paid-work, employment or earnings in Nigeria is very sensitive. We deliberately avoided such questions, especially considering the type of women we interviewed. The tensed situation among the working classes, including incessant strikes by various labour bodies and inaccessibility to the market or farms due to insecurity could distort the intention of this study. However, we could treat this as part of the limitations of this study.
Query: (If the statements were recorded, maybe it is possible, for example, to identify emotions in the surveyed women? This leads to a slightly different way of collecting data - the inclusion of observational data. This is rather a proposition for future research: going beyond the narrative - and content analysis.)
Response: The author has no capacity for emotional analysis but this will be considered in our future research. We will add at least a psychologist to our team to take care of issues like that. We can regard this also as part of the study’s limitation.
Query: The question is whether the title reflects the content of the article. I wonder if I understand the title of the article correctly.
Response: Thanks to the reviewer for alluding to the fact that this research has addressed a problem that is important, interesting and but unfortunately prevalence in many environments. And for agreeing that the study highlighted the limited or narrowing of the system of support for widows. That’s our goal. It means the study has communicated knowledge and has potential for spurring intervention in the respect of what it has highlighted. We appreciate the reviewer for observing that.
Our study only focused on the disposition of in-laws to the widows, especially in terms of exploitation on husband properties. It is based on the culture of parents’ claiming ownership of their son’s asset/property, disregarding the ‘existence’ of his wife or even the son’s children. The culture is so much endemic that the wife could be pushed out of her husband house (or apartment) because ‘whatever the son owns, belongs to his parent’. That is what we have addressed – “in-law interference”.
Logically, while an older adult widow may have gotten children old enough to take care of her (as their mother) or might have built enough resources to sustain herself or could have been so very much integrated into the family or would have made enough friends within the husband family or her community, what befalls a young widow under a few years of marriage experience (<5 years) would certainly be different.
That is what we have addressed. Widow that are Young and of little experience in marriage (<5 years). Majority of them enjoyed good intimacy with their husbands while alive, but immediate the husband dies the in-laws are dispossessing them of their properties or not recognising their rights as wife again within the family. The author believed that exposure of practices like these should be encouraged.
However, except the Reviewer has a title in mind that we should work on, we as author considered the title well explicit enough. What we have addressed will definitely expose further the intricacies in the in-laws’ behaviours and undue interferences in the affairs of couple, not only in Nigeria but in other continents. It will also arouse more support for widows especially the young and those that have short marital encounter (relationship).
